# CenterlineNet: Patch-Aligned Supervision for Thin Road Centerline Extraction

## Abstract

Road networks evolve over time, requiring frequent map updates. AI tools can assist with this task; however, methods based on raster segmentation followed by thinning, skeletonization, or automatic tracing may fail to capture the local structure of road networks, increasing the burden on human annotators. Our goal is to directly predict thin centerline representations that reflect structural patterns used by annotators, particularly at intersections. A secondary goal is to scale training by learning from variable-quality vector data, such as OpenStreetMap, rather than relying on precisely aligned segmentation masks that are difficult to produce at scale. A key challenge is spatial misalignment in training data: while minor for thick segmentation masks, even small shifts become a major obstacle when learning thin centerlines, as pixel-wise losses are disproportionately affected. We propose CenterlineNet, a weakly supervised model that addresses this challenge with a patch alignment loss that compares local neighborhoods instead of individual pixels. This loss matches each predicted neighborhood to its nearest annotated centerline, enabling flexible alignment within a distance tolerance. We present two variants, basic and reciprocal, with the latter handling many-to-one mappings via softmax-in-group weighting, and introduce an intersection-aware component that specifically targets road junctions to improve connectivity.

## 1 Introduction

The extraction of road networks from remote sensing imagery is a fundamental task in computer vision with applications including autonomous driving, urban planning, emergency response, and geographic information systems. Despite advances in deep learning for semantic segmentation, road extraction remains challenging due to the thin, elongated nature of road structures, complex structure relationships, and spatial uncertainties in both imagery and ground truth annotations.

Traditional approaches rely on encoder–decoder architectures such as U-Net (Ronneberger et al., 2015) and DeepLabV3+ (Chen et al., 2018) trained with pixel-wise loss functions like binary cross-entropy or Dice loss. More recent work proposes stronger backbones such as CoANet (Liu et al., 2022) and MSMDFF-Net (Zhang et al., 2023), which incorporate multi-scale context and feature fusion to better handle thin and complex road structures. However, these methods still train with pixel-wise losses that implicitly assume perfect spatial alignment between predictions and labels, an assumption often violated in real-world deployments. Misalignment arises from (1) registration errors between imagery and vector annotations, (2) subjectivity and inconsistency in how annotators interpret road geometry, (3) temporal differences between image acquisition and ground-truth creation, and (4) visual ambiguity or occlusion. Figure 1 illustrates how accurate centerline predictions can disagree with annotated roads under pixel-wise comparison despite capturing the correct structure.

Our contributions are threefold: (1) a patch alignment loss that compares local patches under a label-derived offset field to tolerate spatial offsets while preserving road structure; (2) a reciprocal formulation with softmax-in-group weighting to resolve many-to-one mappings between predictions and ground truth; and (3) an intersection-aware loss component to improve connectivity at road junctions. We demonstrate that CenterlineNet achieves competitive performance on road extraction tasks while remaining robust to annotation noise and spatial misalignments.

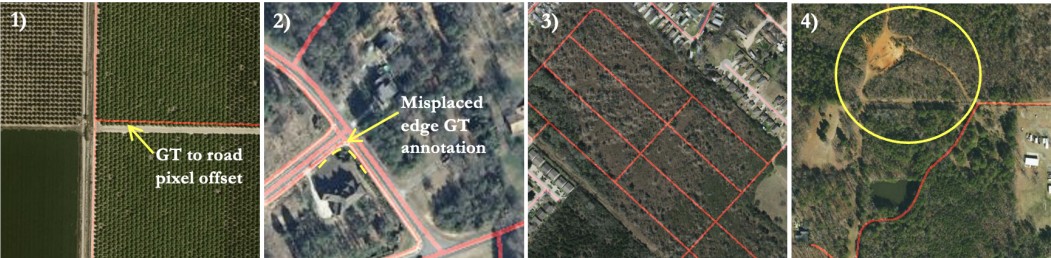

**Figure 1:** Four common road misalignments. **(1)** Ground-truth (GT) pixels displaced from the true road centerline due to registration error; **(2)** annotation variability, GT centerlines include extra edge-line segments; **(3)** temporal mismatches, new roads appear to have been added to forested areas after imagery acquisition; and **(4)** visual ambiguities, roads are missing from GT.

## 2 RELATED WORK

We group related work into three areas relevant to our approach: (1) road extraction using traditional and deep learning, (2) segmentation methods that address spatial misalignment or weak supervision, and (3) approaches designed to preserve connectivity in thin structures. While these research areas provide meaningful advances, they do not jointly address the challenge of learning from spatially uncertain vector-derived centerlines while maintaining the connectivity required for road-network structure.

**Road Extraction.** Early road extraction relied on handcrafted features, edge detection, and morphological operators (Mena, 2003; Hu et al., 2007). Deep learning methods later shifted toward encoder–decoder architectures such as U-Net (Ronneberger et al., 2015) and DeepLab (Chen et al., 2018), followed by road-specific variants including Deep Residual U-Net (Zhang et al., 2018), D-LinkNet (Zhou et al., 2018), and structural-similarity–guided designs (He et al., 2019b).

Beyond pixel-based segmentation, several models incorporate more explicit structural reasoning. RoadTracer (Bastani et al., 2018) reframes road extraction as an iterative graph-expansion process that directly predicts topological connectivity, but this is different than our semantic segmentation approach and isn't tested against poorly annotated data. CoANet (Liu et al., 2022) improves connectivity by embedding structural cues into the feature hierarchy, yet it still assumes accurate pixel-level annotations. More recent transformer-augmented approaches such as Seg-Road (SAMRoad) (Tao et al., 2023) integrate global attention with CNN features to better capture long-range road structure, although they similarly depend on precise training masks.

Transformer-based road extraction such as RoadFormer (Rao & et al., 2023) and hybrid raster–vector mapping frameworks including VectorMapNet (Zhu et al., 2023) further improve global context and geometric fidelity. Multi-scale and multi-directional feature fusion has also been explored for road delineation (Zhang et al., 2023). Despite these advances, these models rely on well-aligned pixel-level supervision, making them sensitive to the misregistration commonly found in vector-derived centerline labels.

**Weak Supervision and Spatial Misalignment.** Several works attempt to make segmentation tolerant to spatial noise. Batra et al. combine segmentation with orientation cues and post-processing to reinforce connectivity (Batra et al., 2019), but their formulation remains tied to thick pixel-aligned masks. Other strategies indirectly tolerate misalignment through relaxed evaluation metrics or buffered targets (Sun et al., 2019), though these adjustments occur after training and do not modify the loss itself.

A separate direction improves the quality of the labels. STEAL (Acuna et al., 2019) uses edge-aware self-training to refine noisy annotations, and alignment pipelines such as Zhang et al. (Zhang et al., 2020) attempt to correct OSM-derived labels before supervision. While these approaches reduce coarse annotation noise, they do not make the training objective intrinsically tolerant to the small spatial shifts that occur when rasterizing thin vector centerlines.

More recently, weakly supervised segmentation has explored uncertainty modeling (Li & Chen, 2023), self-supervised pseudo-label denoising (Wu & et al., 2024), and spatial uncertainty modeling

for shifted masks (Choe & Ji, 2024). These strategies improve robustness to noisy or incomplete masks, but they are designed for region-level semantics. They do not address the sensitivity of one-pixel-wide structures, where even minor misalignment can invert the supervision signal.

**Structural Consistency in Thin Structures.** A complementary line of work emphasizes topological correctness. Persistent-homology-based losses (Mosinska et al., 2018) and connectivity-focused objectives such as GapLoss (Yuan & Xu, 2022) explicitly encourage continuity in predicted structures. Centerline-focused approaches include two-stage segmentation-and-tracing pipelines (Wei et al., 2020) and joint road-surface/centerline extraction such as MRENet (Lu et al., 2021).

Connectivity-preserving objectives have recently been strengthened via Skeleton Recall Loss (Kirchhoff et al., 2024), differentiable connectivity constraints in DiffConnect (Chen & et al., 2023), and topology-consistent segmentation under geometric distortions (Yang & et al., 2024). Related representation-learning approaches leverage orientation cues (Kim & et al., 2023), skeleton-guided alignment (Xu & et al., 2023), curvature-aware representations (Han & et al., 2024), and shape-aligned feature regularization (Fan & et al., 2024) to enhance structural fidelity. However, all these methods assume aligned pixel-level labels; therefore, structurally correct but spatially shifted predictions are still penalized under misregistered supervision.

## 3 METHODOLOGY

The accurate extraction of road centerlines from remote sensing imagery remains a complex challenge due to the presence of spatial misalignments and the intricate structure of road networks. To address these issues, we introduce a loss formulation tailored for singleton centerline localization, where "singleton" denotes the nearly one-pixel-wide centerline representation used as ground truth rather than thick road polygons. The following section outlines the methodological framework, presenting the network design, loss functions, and training strategies employed in our approach.

### 3.1 DEEPLABUNETPRECISE ARCHITECTURE

CenterlineNet uses a hybrid backbone, which we call DeepLabUNetPrecise. The encoder follows DeepLabV3+, using a ResNet-101 with Atrous Spatial Pyramid Pooling (ASPP) to capture multi-scale context. Unlike prior hybrid models (He et al., 2019a; Zhou et al., 2018) that pool to $1/16$ or $1/32$ scale, our encoder stops at $1/8$ resolution and substitutes atrous convolutions for deeper pooling. This preserves finer feature maps while still expanding the receptive field. The decoder then upsamples through U-Net–style skip connections to restore predictions at full input resolution, which is essential for reconstructing one-pixel-wide road centerlines.

The architecture itself follows established hybrid patterns; our main contribution lies in the loss design. The role of DeepLabUNetPrecise is to provide sufficient resolution and feature detail so that the proposed patch alignment and intersection-aware losses can operate effectively.

### 3.2 LOSS FUNCTIONS

Our central idea is a patch alignment loss that tolerates spatial misalignment by comparing small neighborhoods rather than individual pixels. For each prediction location, we align its predicted logit-patch (logits of the $K$ nearest pixels centered on a prediction) to the most relevant ground-truth neighborhood and measure per-patch cross-entropy. This preserves thin structures while allowing small spatial shifts.

For every pixel $\mathbf{x} = (x, y)$, we compute an offset vector $\mathbf{v}(\mathbf{x})$ that points to the nearest ground-truth centerline pixel. This offset field serves two roles: (i) it defines a tolerance band around the centerline (used later via a binary mask to include/exclude locations), and (ii) it provides the correspondence needed to extract and compare a predicted patch $\widehat{\mathbf{p}}(\mathbf{x})$ with the ground-truth patch $\mathbf{p}(\mathbf{x} + \mathbf{v}(\mathbf{x}))$. The exact loss expressions and masks are given in the subsections that follow.

We train CenterlineNet with a weighted sum of four terms:

$$\mathcal{L}_{\text{total}} = \mathcal{L}_{\text{patch}} + \alpha \, \mathcal{L}_{\text{fp}} + \beta \, \mathcal{L}_{\text{singleton}} + \gamma \, \mathcal{L}_{\text{intersection}}, \tag{1}$$

where $\alpha = 5.0$, $\beta = 0.5$, and $\gamma = 1.0$ in our experiments. Each term is defined below.

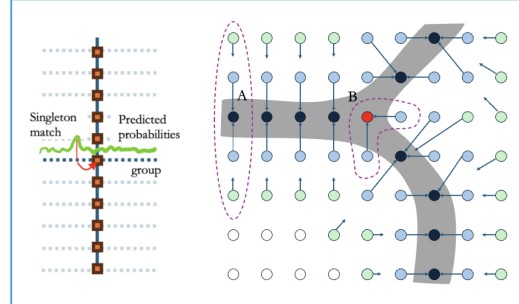

**Figure 2: Patch alignment and reciprocal grouping.** Left: Patch alignment aligns each predicted pixel to its nearest point on the ground-truth centerline and extracts small, spatially corresponding patches around both locations. The values shown are examples the model's log-probabilities, and loss is computed by comparing them pixel-by-pixel with a patchwise cross-entropy that enforces local geometric structure rather than relying on a single-pixel match. Right: Reciprocal grouping addresses the common many-to-one problem, where several predicted pixels map to the same ground-truth point. All such predictions form a group, and we apply a softmax over their logits to weight the contribution of each prediction. The loss is then concentrated on the most confident pixel in the group, while weaker or redundant predictions are suppressed. This produces sharper centerlines and stabilizes training in cluttered or ambiguous regions.

### 3.2.1 RECIPROCAL SOFTMAX-IN-GROUP WEIGHTING

To resolve many-to-one correspondences, each pixel $\mathbf{x}$ that maps to the same ground-truth location as others is assigned a reciprocal weight $w_{\mathbf{x}}$. Let $G(\mathbf{x})$ denote this group of pixels and $\tau$ a temperature parameter that controls the sharpness of the distribution. The weights are defined by a softmax over logits within each group:

$$w_{\mathbf{x}} = \frac{\exp\left(\frac{z(\mathbf{x})}{\tau}\right)}{\sum_{\mathbf{y} \in G(\mathbf{x})} \exp\left(\frac{z(\mathbf{y})}{\tau}\right)}. \tag{2}$$

This ensures that only the most confident prediction in each group receives high weight, reducing ambiguity when multiple pixels compete for the same ground-truth location.

### 3.2.2 WEIGHTED PATCH ALIGNMENT LOSS

First, we define the patch alignment loss

$$\mathcal{L}_{\text{patch}} = \frac{\sum_{\mathbf{x}} m(\mathbf{x}) \, w_{\mathbf{x}} \, \text{CE}\big(\widehat{\mathbf{p}}(\mathbf{x}), \mathbf{p}(\mathbf{x} + \mathbf{v}(\mathbf{x}))\big)}{\left(\sum_{\mathbf{x}} m(\mathbf{x}) \, w_{\mathbf{x}}\right) \cdot K}. \tag{3}$$

Here $CE(\cdot, \cdot)$ is the binary cross-entropy on logits, $m(\mathbf{x})$ is a binary mask indicating whether pixel $\mathbf{x}$ lies within distance $d_{\max}$ (a hyperparameter set to double the average road-width) of a ground-truth centerline, $w_{\mathbf{x}}$ is the reciprocal softmax weight (defined in eq 2), $\mathbf{v}(\mathbf{x})$ is the offset vector at pixel $\mathbf{x}$, $\widehat{\mathbf{p}}(\mathbf{x})$ is the set of logits in the predicted patch, $\mathbf{p}(\mathbf{x} + \mathbf{v}(\mathbf{x}))$ the binary labels of the corresponding ground-truth patch, and $K$ the number of pixels in a patch. The loss averages binary cross-entropy errors across masked patches, with weights emphasizing high-confidence predictions and normalization ensuring comparability across patches.

### 3.2.3 FALSE POSITIVE LOSS

The false-positive loss penalizes road predictions farther than $d_{\max}$ from any ground-truth centerline:

$$\mathcal{L}_{\text{fp}} = \frac{1}{|\Omega|} \sum_{\mathbf{x} \in \Omega} \big(1 - m(\mathbf{x})\big) \operatorname{CE}(z(\mathbf{x}), 0), \tag{4}$$

where $\Omega$ is the image domain, $m(\mathbf{x}) \in \{0, 1\}$ is the validity mask ($m(\mathbf{x}) = 1$ inside the tolerance band and 0 otherwise), and $z(\mathbf{x})$ is the logit at pixel $\mathbf{x}$. This loss suppresses predictions outside the valid road region.

### 3.2.4 SINGLETON LOSS

The singleton loss $\mathcal{L}_{\text{singleton}}$ ensures that only one pixel in each group of predictions is activated. For each pixel $\mathbf{x}$, let $G(\mathbf{x})$ denote the set of predictions whose offset vectors map them to the same ground-truth location. We then assign binary targets $t_{\mathbf{x}}$ so that exactly one pixel in each group receives $t_{\mathbf{x}} = 1$ (the one with the highest logit, ties broken by proximity to the ground truth) and all others receive $t_{\mathbf{x}} = 0$. These targets are derived deterministically and do not need to be differentiable. Then

$$\mathcal{L}_{\text{singleton}} = \frac{1}{\sum_{\mathbf{x}} m(\mathbf{x})} \sum_{\mathbf{x}} m(\mathbf{x}) \operatorname{CE}(z(\mathbf{x}), t_{\mathbf{x}}), \tag{5}$$

which encourages the network to concentrate confidence on a single representative pixel per ground-truth location, reducing redundant predictions and focusing on a single, thin line prediction.

### 3.3 INTERSECTION-AWARE LOSS

The intersection loss $\mathcal{L}_{\text{intersection}}$ enforces structural connectivity at road junctions, where prediction errors are especially costly. We detect intersections as road pixels that have more than two neighbors in a designated window. Let $N_{\text{int}}$ denote the number of such ground-truth intersections, and let $\mathbf{p}(\mathbf{x}_i)$ be the ground-truth patch centered at intersection $i$. For each intersection, we search within a radius $r$ for the predicted patch $\widehat{\mathbf{p}}(\mathbf{x}_j)$ that minimizes cross-entropy with the ground truth:

$$\mathcal{L}_{\text{intersection}} = \frac{1}{N_{\text{int}}} \sum_{i=1}^{N_{\text{int}}} \min_{\|\mathbf{x}_j - \mathbf{x}_i\| \leq r} \operatorname{CE}(\widehat{\mathbf{p}}(\mathbf{x}_j), \mathbf{p}(\mathbf{x}_i)). \tag{6}$$

At junctions, multiple road branches meet, and the offset field becomes unstable, so nearest-neighbor assignment based on the vector field $\mathbf{v}(x)$ may select the wrong prediction. To avoid this, we treat all predicted points within a search radius as potential matches to an intersection, and supervise them jointly against the ground-truth patch, ignoring the vector field. This collective supervision maintains connectivity and yields more reliable learning at intersections.

## 4 EXPERIMENTAL EVALUATION

We conduct comprehensive experiments to evaluate CenterlineNet's performance on road extraction and intersection topology preservation. Our evaluation addresses the fundamental challenges of assessing methods designed to handle spatial misalignment by utilizing specialized metrics that focus on topological accuracy rather than pixel-perfect correspondence.

### 4.1 EVALUATION METHODOLOGY

The evaluation of road extraction methods poses fundamental challenges when spatial misalignment is present. Traditional pixel-wise metrics such as Intersection over Union (IoU), precision, and recall operate under the assumption of perfect spatial correspondence between predictions and ground truth annotations. This assumption directly contradicts our method's core premise of handling spatial uncertainties and misalignments that are inherent in real-world remote sensing data.

To address this contradiction, we develop a comprehensive evaluation framework that measures accuracy and structural preservation while accounting for reasonable spatial tolerances. Our approach

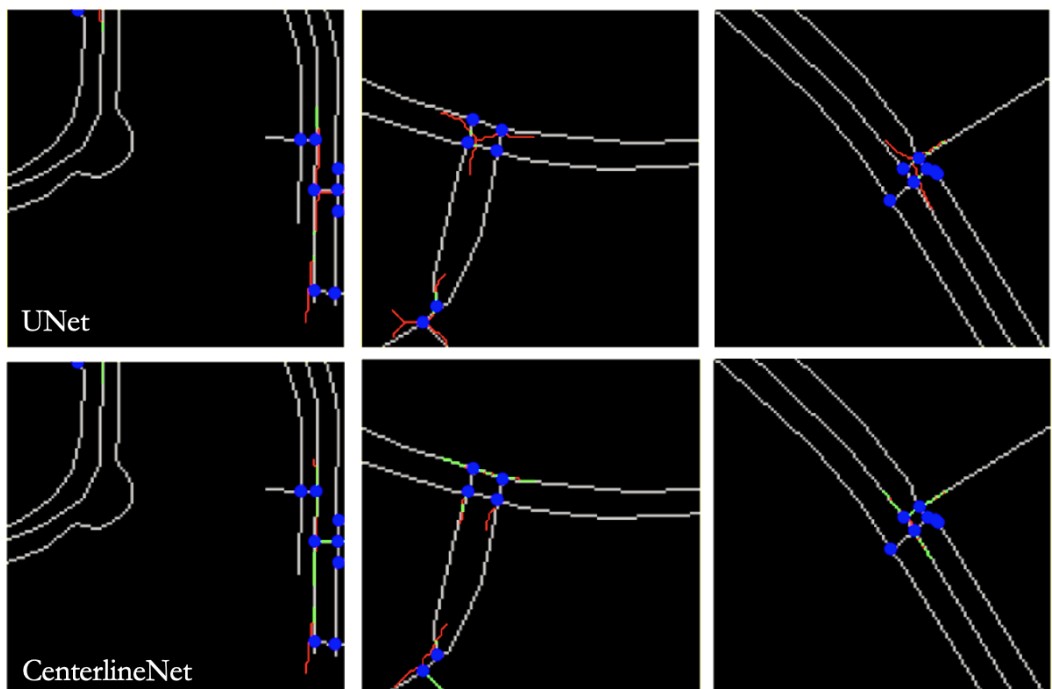

**Figure 3: Intersection detection and targeted supervision.** Intersections (blue dot) with U-Net+CE (top row) vs. CenterlineNet (bottom row). Improved intersection awareness is shown with green overlays (prediction aligned with ground truth) for CenterlineNet, versus red overlays (prediction outside tolerance) for the baseline.

recognizes that for road network extraction, the preservation of connectivity, intersection structure, and overall network structure is more critical than pixel-perfect alignment.

Our primary evaluation approach employs skeleton-based bipartite matching to assess road network quality independent of minor spatial offsets. Both predicted road networks and ground truth annotations undergo morphological thinning operations to produce nearly one-pixel-wide centerline representations. This preprocessing step removes large variations in road width prediction and focuses evaluation on the fundamental network structure and connectivity patterns.

We establish correspondences between predicted and ground truth skeleton pixels using the Hungarian algorithm for optimal bipartite matching. The matching process operates within varied distance threshold of pixels, which reflects realistic spatial tolerances for satellite imagery while accounting for typical registration errors, annotation inconsistencies, and the inherent uncertainties in manual road labeling.

### 4.2 Intersection Structure Evaluation

Road intersections represent critical structural features that define network connectivity and determine practical utility for navigation applications. Unlike road segments, intersections present unique evaluation challenges due to their small spatial extent, complex geometric structure, and high importance for overall network functionality. We detect intersections in skeletonized road networks as pixels in the ground truth having more than two neighbors within their local morphological neighborhood. For evaluation, we employ a patch-based approach that directly compares the local road structure around intersection locations.

For each ground truth intersection, we extract a local pixel patch centered at the intersection location from the ground truth skeleton. We then find the spatially closest predicted intersection within a predetermined radius and extract a corresponding patch from the predicted skeleton. The matching is based purely on Euclidean distance between intersection centers, ensuring that the subsequent patch

Table 1: **Quantitative results on Centerline1M.** Evaluation on skeletonized predictions and ground-truth centerlines. Metrics reported include bipartite-matching precision/recall/F1 (3-px tolerance) and structural IoU.

| Model | Loss | Precision ↑ | Recall ↑ | F1 ↑ | Struct. IoU ↑ |
|---|---|---|---|---|---|
| U-Net | CE | 0.944 | 0.787 | 0.858 | 0.134 |
| U-Net | Combined | 0.938 | 0.807 | 0.867 | 0.123 |
| DeepLabV3 | CE | 0.974 | 0.771 | 0.861 | 0.154 |
| DeepLabV3 | Dice | 0.989 | 0.766 | 0.864 | 0.185 |
| ResNet-FPN | CE | 0.982 | 0.754 | 0.853 | 0.146 |
| SAMRoad | CE | 0.988 | 0.308 | 0.450 | 0.090 |
| CenterlineNet | CE | **0.999** | 0.737 | 0.849 | 0.090 |
| CenterlineNet | CE + Patch | 0.982 | 0.780 | 0.869 | 0.149 |
| CenterlineNet | CE + Patch + Reciprocal | 0.993 | **0.793** | **0.878** | **0.160** |
| CenterlineNet | CE + Patch + Intersection | 0.993 | 0.782 | 0.871 | 0.149 |
| CenterlineNet | Combined | 0.978 | 0.783 | 0.870 | 0.155 |

IoU evaluation provides an unbiased measure of structural similarity. If no predicted intersection exists within the specified radius, the ground truth intersection is considered unmatched for recall calculation.

The intersection quality is evaluated using the Intersection over Union (IoU) between these spatially matched patches, which captures both the spatial accuracy of the intersection location and the preservation of the local road network structure. This patch-based evaluation measures fundamentally different aspects of intersection quality compared to the bipartite matching approach used for road segments.

While bipartite matching focuses on whether individual skeleton pixels can be paired within a distance threshold, the patch-based IoU captures the local geometric shape and structure of intersections. It evaluates whether the predicted intersection preserves the correct angular relationships between incident road segments, maintains proper connectivity patterns, and reproduces the overall geometric configuration of the intersection. For example, a T-junction with specific arm angles and lengths will have a characteristic patch pattern that must be preserved to achieve high IoU scores.

The predetermined patch size is large enough to capture the full intersection structure including the immediate approach segments, while remaining focused on the local neighborhood. This approach naturally handles variations in intersection geometry, missing or extra road segments, and small spatial offsets while providing a direct measure of how well the predicted network preserves the local connectivity structure and geometric shape around intersections. The evaluation encompasses structural IoU (average IoU of matched intersection patches), results shown in Table 2.

## 4.3 EXPERIMENTAL SETUP AND DATASETS

All experiments were run on two NVIDIA TITAN RTX GPUs (24 GB each). We trained CenterlineNet on **Centerline1M**, our 1 m-resolution dataset of U.S. road centerlines automatically derived from USGS imagery (M2M API) and OpenStreetMap. We accessed the computational expense of utilizing our CenterlineNet model with traditional loss functions versus our novel loss function contributions and saw a minimal 2.0% increase in runtime and 12.5% increase in GPU usage. While our model was running on a batch size of 8 we have the capabilities to run a batch size of 16. Centerline1M intentionally retains the noisy, misaligned nature of crowd-sourced labels: OpenStreetMap vectors were rasterized into nearly one-pixel binary masks (anti-aliasing off), keeping only drivable road types (e.g., `highway=primary`, `secondary`, `residential`, `tertiary`) and ignoring service and non-drivable classes. No manual correction or alignment refinement was applied. The dataset comprises 10,845 tiles (8,579 training, 2,266 validation).

To test generalization, we evaluated on a well-known dataset. **SpaceNet Roads**[1] offers multi-city, multi-resolution satellite imagery together with road vector annotations (centerline graphs), providing a complementary test of geographic and sensor transfer (Van Etten et al., 2018). We rasterized and skeletonized these vector road networks in order to align with our own evaluation pipeline.

---

[1]`https://spacenet.ai/datasets/`

Additionally, we tested against **RoadTracer**, which combines aerial imagery with OpenStreetMap road graphs and evaluates performance using a graph-based junction metric (Bastani et al., 2018).

**Table 2: Quantitative Results (SpaceNet and RoadTracer):** Bipartite evaluation metrics (3-px tolerance) comparing CoANet and CenterlineNet.

| Dataset | Model | Precision ↑ | Recall ↑ | F1 ↑ |
|---------|-------|-------------|----------|------|
| **SpaceNet** | CoANet | 0.521 | **0.501** | 0.492 |
| | CenterlineNet | **0.546** | 0.465 | **0.624** |
| **RoadTracer** | CoANet | 0.189 | 0.195 | 0.167 |
| | CenterlineNet | **0.603** | **0.419** | **0.744** |

## 4.4 OCCLUSIONS

A critical challenge in road centerline extraction from satellite imagery is the presence of occlusions caused by trees, buildings, shadows, or cloud cover, which obscure parts of the road network. These occlusions introduce gaps in visual continuity and spatial misalignment between predictions and ground truth. To improve resilience under these conditions, we augment our training data with synthetic occlusions in addition to naturally occluded scenes.

For each training tile we randomly select a road pixel from the ground-truth mask (or a random location if no road pixel exists) and center a rectangular occlusion over that point. The occlusion size is sampled between 0.5× and 1.0× of a base fraction of the image dimensions (0.25 in our experiments), producing variable occlusion shapes. Within this rectangle we replace the original pixel values with the mean color of the entire image rather than black or random noise. The same occlusion rectangle is applied to all input channels, and we also generate a binary occlusion mask to record the affected area. Ground-truth road masks themselves are left unaltered so that the model still receives supervision at occluded locations.

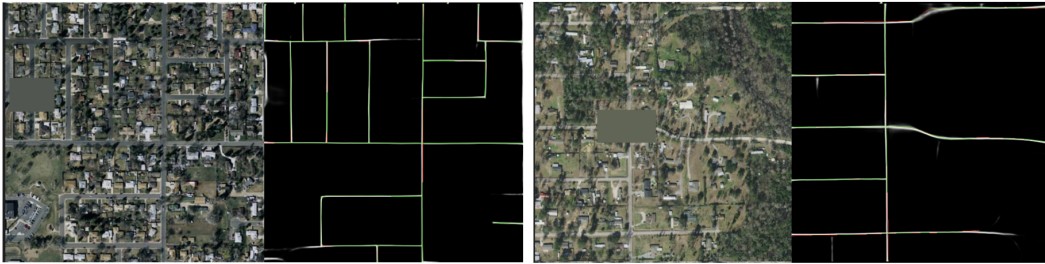

**Figure 4:** Examples of CenterlineNet predictions on synthetically occluded satellite imagery. For each scene, the left panel shows the original satellite image with an artificial occlusion (gray rectangle), and the right panel shows the corresponding CenterlineNet prediction overlaid on a black background. Green pixels indicate predicted road centerlines correctly matching ground truth within the spatial tolerance, while red pixels mark mismatches. Additional support image predictions shown in Supplemental Material Fig. 7.

This augmentation forces CenterlineNet to infer road connectivity across missing visual segments, reduces false negatives in occluded regions, and improves generalization to deployment scenarios where occlusion is the norm rather than the exception.

## 4.5 RESULTS AND ANALYSIS

CenterlineNet achieves competitive performance compared to baseline methods across overall road extraction metrics, with improved Precision, F1, and Structural IoU metric scores. Not only quantitatively we can see improvements qualitatively see Figure 5. In UNet and DeepLab predictions we can see not structurally correct prediction branches, even though DeepLab with Dice loss seems to have success over its predecessors CenterlineNet still shows optimal improvements over the later.

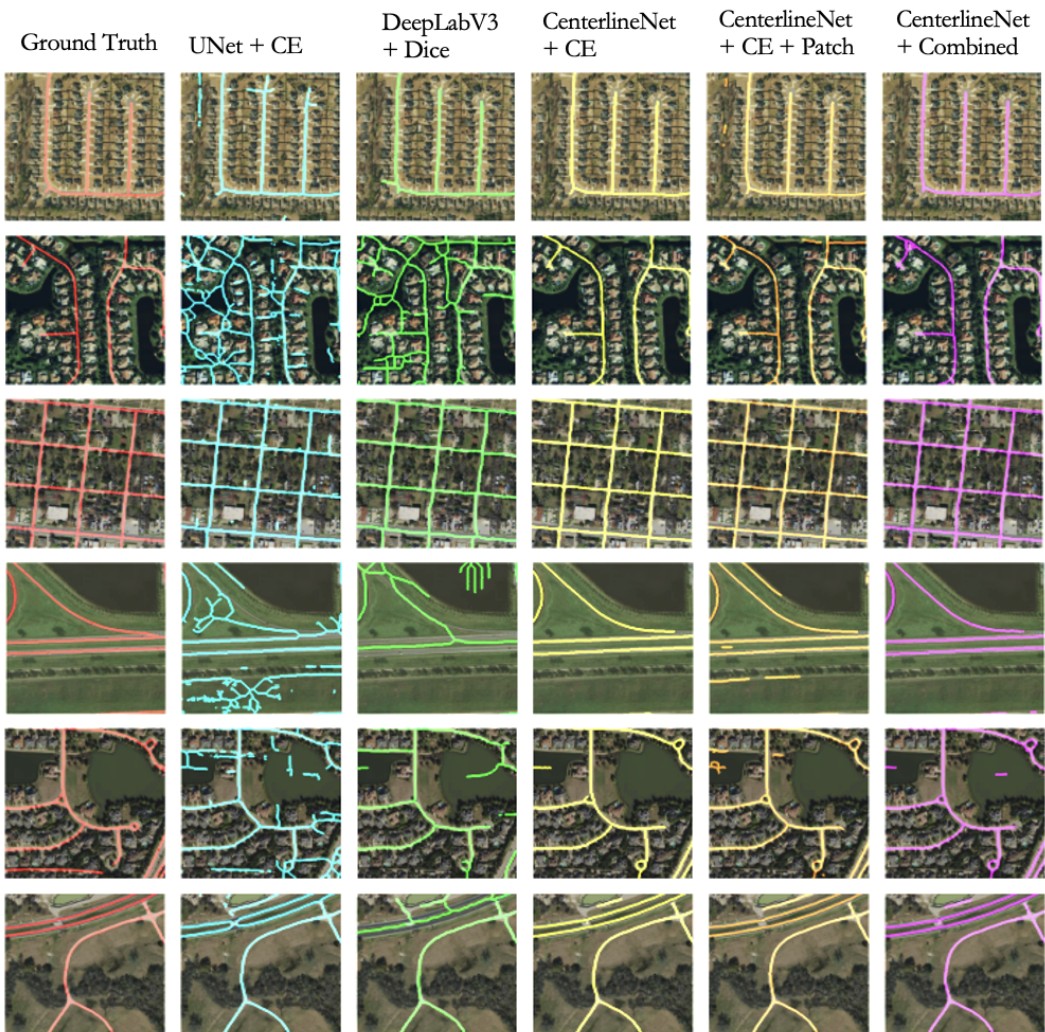

**Figure 5: Qualitative Results:** Aerial images overlayed with thickened groundtruth and various model thickened predictions masks showing improved structure accuracy with CenterlineNet.

To better understand our best models metrics we have taken and implemented our bipartite matching metrics not only on road feature types that are drivable but on other OpenStreetMap types such as paths, raceways, footways, and others. Table 3 shows the break down of Precision, Recall and F1 scores for our best model based on various feature types. These results show that in drivable road feature types our model has high metric values and struggles some with poorly annotated feature types that such as paths or construction roads.

We conduct an ablation studies to understand the contribution of each loss component in our approach. We evaluate a baseline configuration using standard binary cross-entropy loss with pixel-wise alignment assumptions, then systematically add the patch alignment loss while maintaining standard assumptions for other components. We further examine the contribution of softmax-in-group weighting to handle many-to-one mapping scenarios, and finally integrate specialized supervision for intersection detection and preservation.

As shown in Table 1, the inclusion of different components leads to varied improvements across Precision and F1 scores. Notably, the patch alignment loss substantially boosts Precision and F1, while the reciprocal formulation improves Recall. The addition of intersection supervision further enhances precision and F1, demonstrating the complementary benefits of various loss combinations.

**Table 3:** Per-class performance across OSM highway tags for CenterlineNet Model. Metrics include precision (P), recall (R), and F1 score.

| OSM Highway Tag | P ↑ | R ↑ | F1 ↑ |
|---|---|---|---|
| primary | 0.988 | 0.933 | 0.960 |
| trunk_link | 0.995 | 0.891 | 0.940 |
| motorway | 0.984 | 0.877 | 0.928 |
| tertiary | 0.987 | 0.827 | 0.900 |
| secondary | 0.966 | 0.784 | 0.867 |
| residential | 0.986 | 0.769 | 0.864 |
| raceway | 0.935 | 0.712 | 0.809 |
| trunk | 0.966 | 0.641 | 0.771 |
| unclassified | 0.995 | 0.583 | 0.734 |
| service | 0.979 | 0.527 | 0.685 |
| motorway_link | 0.965 | 0.418 | 0.583 |
| cycleway | 0.988 | 0.401 | 0.571 |
| footway | 0.984 | 0.322 | 0.486 |
| track | 0.990 | 0.244 | 0.392 |
| construction | 0.957 | 0.198 | 0.329 |
| secondary_link | 0.897 | 0.153 | 0.261 |
| path | 0.993 | 0.739 | 0.138 |

Despite its improvements, CenterlineNet sometimes struggles with occluded roads, very thin or low-contrast rural roads, and complex highway interchanges, see supplementary materials.

## 5 CONCLUSION

We presented CenterlineNet, a weakly supervised approach for road centerline extraction that addresses spatial misalignment in remote sensing applications. Our patch alignment loss provides spatial tolerance while maintaining topological accuracy, suitable for real-world scenarios where perfect annotation alignment cannot be guaranteed. Our contributions include: (1) a patch alignment loss using vector fields to establish flexible correspondences, (2) a reciprocal formulation handling many-to-one mappings through softmax-in-group weighting, and (3) an intersection-aware component improving network connectivity. Results demonstrate competitive performance with improved robustness to spatial noise and annotation inconsistencies. The approach enables practical deployment where spatial uncertainties are inherent. Future work will explore extension to other linear infrastructure extraction tasks and integration with vector post-processing methods.

ACKNOWLEDGMENTS

AI tools, ChatGPT and GitHub Copilot, were used during polishing writing and researching similar model publications for prior art. The authors directed the work, made all research decisions, and carried out the analysis with substantial human effort.

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

## A    APPENDIX

This section presents additional examples of CenterlineNet in situations involving occlusions and challenges common to aerial and satellite imagery. The main paper showed skeletonized results – but we wanted to show that our pre-thinned results are already very thin. We provide both skeletonized and non-skeletonized visualizations to demonstrate how our patch-reciprocal-intersection loss function handles predictions under an added occlusion. In each of the images, a random rectangle is replaced by the mean color of the satelite image.

### A.1    CENTERLINENET
### WITH PATCH-RECIPROCAL-INTERSECTION

Fig. 7 (a full-page figure) shows centerlines extracted by our model for numerous examples. Our quantitative evaluation first skeletonizes and then matches pixels to ground-truth, but that does not capture the fact that our predictions are already very thin. In these figures, no bipartite matching is done, we simply compare the ground-truth the thresholded prediction (with a threshold of 50%). Green pixels are true positives for predictions, while red pixels are false positives. We direct the reader's attention to these observations:

- **Occlusion fill.** The network bridges many occluded gaps, though with lower certainty, visible as slightly brighter blobs in the background pixels. This is most visible in row 2 image set 1, and to some extent in row 3 image set 2

- **Split-lane awareness.** On divided highways such as row 5 set 1 marked as B with red arrow, the model outputs two parallel centerlines. It hallucinates plausible two-lane roads in the occluded area in row 6 image set 1.

- **Plausible hallucination.** When detail is completely hidden, the network invents reasonable geometry (e.g., row 6 image set 2); we regard this as a feature rather than a flaw.

- **Minor structures.** Driveways and access roads appear inconsistently, for example row 6 image set 2 in the bottom right quadrant, mirroring the uneven annotation quality in OpenStreetMap.

## B    ADDITIONAL EXPERIMENTAL RESULTS

Table 5 compares model variants trained with and without occlusion. Included metrics Precision, Recall, and F1 scores. Models trained with occlusion show clear improvement, especially in complex junction detection, highlighting the benefit of occlusion-aware training.

## C    FAILURE AND ABNORMAL CASES

Fig. 8 illustrates selected examples where CenterlineNet encounters difficulties in accurately extracting road centerlines. We observe the following patterns:

1. Occlusion near the edge of an image is more damaging - presumably, because there is not as much context.

2. Occasionally, roads are missing in our validation set, but are found by the model. These are not true failures.

3. Excessive shadows or forests that occlude the road cause failures. There is a limit to how much occlusion we can handle. This is similar to occlusion near the edges - there is not enough context to predict the occluded region.

## D    NOTATION AND HYPER-PARAMETERS

Table 4 summarizes the variables and hyperparameters used in our loss functions and best saved model.

# E PIXELWISE TOLERANCE ABLATION STUDY

Fig. 6 shows various pixel spatial tolerances used to to determine the practically of pixel spatial tolerance selection. Real-world applications may require understanding models across a range of tolerances. The figure reports CenterlineNet's precision, recall, and F1 scores as a function of the pixel matching threshold base on Centerline1M dataset. These results demonstrate how spatial tolerance parameters impact both network extraction and connectivity evaluation under varying annotation registration conditions. Sharp incline in scores from 1 to 3 pixel tolerance but not a significant impact in score increase from 3 to 15 pixels.

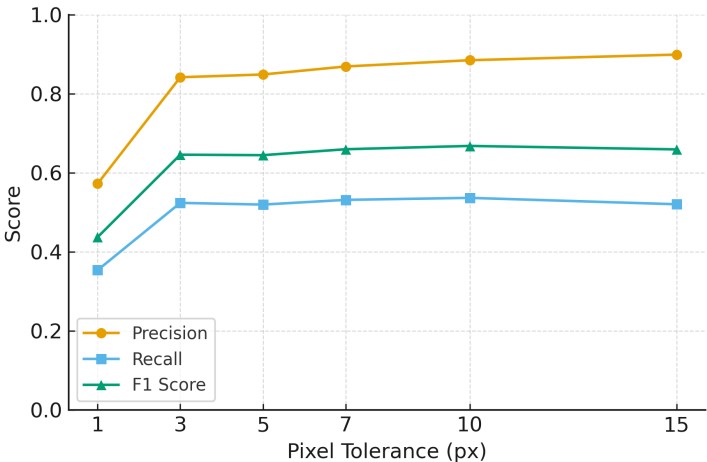

**Figure 6:** Precision, recall, and F1 scores of CenterlineNet as a function of pixel-matching tolerance.

**Table 4:** Complete Training and Evaluation Implementation Details. All CenterlineNet variants are implemented using the DeepLabUNetPrecise backbone with 512×512 inputs. Patch, Reciprocal, and Intersection correspond to the three components of our proposed loss.

| Component | Specification |
|---|---|
| **Dataset and Preprocessing** | |
| Dataset | USGS–OSM satellite imagery crops (512×512) |
| Resolution | 1 meter/pixel |
| Training Samples | 512×512 pixel patches extracted from large aerial scenes |
| Normalization | ImageNet mean/std: (0.485, 0.456, 0.406) / (0.229, 0.224, 0.225) |
| **Network Architecture** | |
| Backbones | UNet, DeepLabUNetPrecise, DeepLabv3+, ResNet50+FPN |
| Input Size | 512×512×3 |
| Output Channels | 1 (binary centerline mask) |
| **Training Hyperparameters** | |
| Optimizer | Adam |
| Learning Rate | 5e-5 |
| Weight Decay | 1e-4 |
| Batch Size | 8–16 depending on model and GPU memory |
| Epochs | 100 with early stopping |
| Loss Function | BCE (roads) + Patch Alignment + Reciprocal + Intersection Loss |
| **Patch Alignment Loss Parameters** | |
| Patch Size | 11×11 pixels |
| Search Radius | 5 pixels |
| $\alpha$ (False Positive Weight) | 5.0 |
| $\beta$ (Singleton Weight) | 0.5 |
| **Hardware and Runtime** | |
| Hardware Used | NVIDIA TITAN RTX (24GB) and GTX 1080Ti (11GB) |
| Framework | PyTorch 1.x |
| CUDA Version | 11.x |
| Training Time | 4–8 hours per model (100 epochs, depending on variant) |
| GPU Memory Usage | 8–15GB depending on loss components |

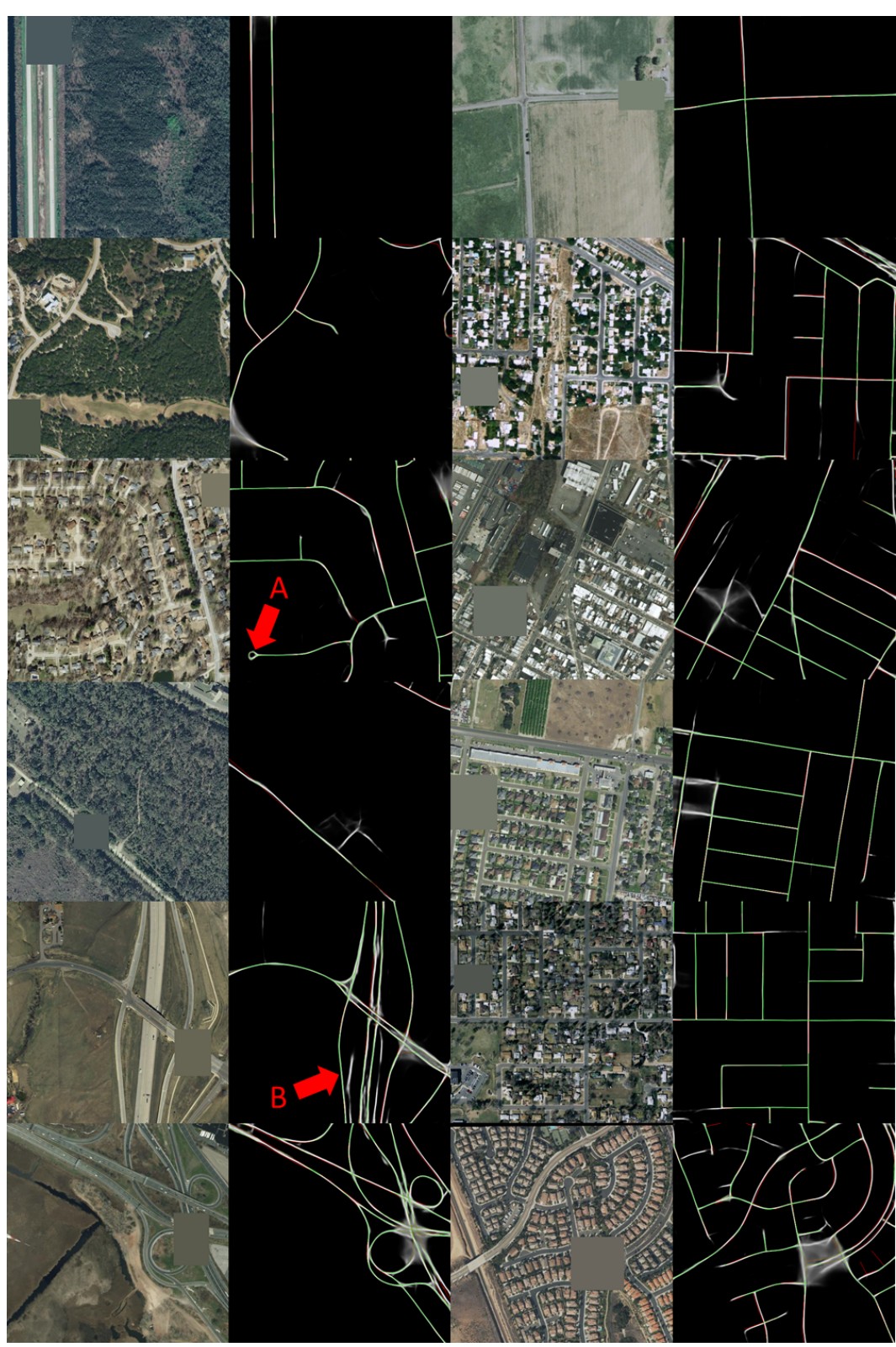

**Figure 7:** Qualitative results of occluded predictions using CenterlineNet

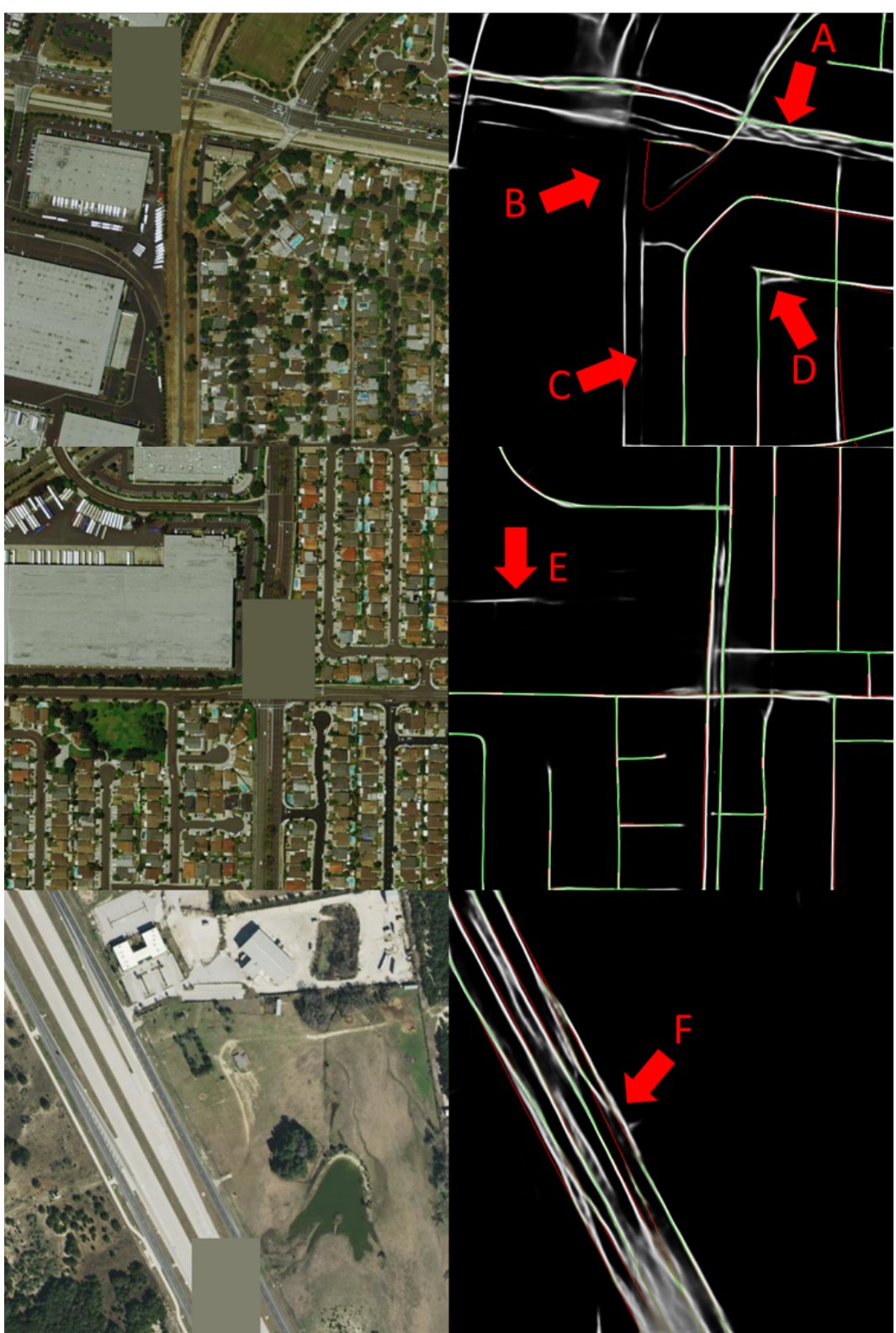

**Figure 8:** Failure prediction cases

**Table 5:** Bipartite Matching Metrics Grouped by Method and Occlusion.
We **bold** the best score in each row, and in case of a tie both are bolded. The second-best are
underlined.

| Valence & Metric | UNet | | Patch | | Patch-Reciprocal | | Patch-Recip-Intersect | |
|---|---|---|---|---|---|---|---|---|
| | Non-Occ | Occ | Non-Occ | Occ | Non-Occ | Occ | Non-Occ | Occ |
| **Bipartite Matching Metrics** | | | | | | | | |
| Overall Precision | 0.51 | 0.51 | 0.29 | **0.82** | 0.27 | 0.67 | 0.28 | 0.81 |
| Overall Recall | 0.50 | 0.26 | 0.47 | 0.49 | 0.45 | **0.54** | 0.43 | 0.49 |
| Overall F1 Score | 0.51 | 0.37 | 0.36 | **0.61** | 0.34 | 0.60 | 0.34 | **0.61** |
| **End (Valence 1)** | | | | | | | | |
| Precision | 0.02 | 0.00 | 0.01 | **0.18** | 0.00 | 0.04 | 0.00 | **0.18** |
| Recall | 0.07 | 0.00 | 0.02 | 0.23 | 0.01 | 0.15 | 0.02 | **0.24** |
| F1 Score | 0.03 | 0.00 | 0.01 | 0.20 | 0.01 | 0.07 | 0.01 | **0.21** |
| **Straight (Valence 2)** | | | | | | | | |
| Precision | 0.39 | 0.06 | 0.21 | **0.65** | 0.21 | 0.42 | 0.20 | **0.65** |
| Recall | 0.43 | 0.13 | 0.38 | 0.45 | 0.36 | **0.49** | 0.36 | 0.46 |
| F1 Score | 0.41 | 0.09 | 0.28 | **0.53** | 0.26 | 0.46 | 0.26 | **0.53** |
| **T (Valence 3)** | | | | | | | | |
| Precision | 0.04 | 0.00 | 0.01 | 0.25 | 0.01 | 0.07 | 0.01 | **0.26** |
| Recall | 0.23 | **0.32** | 0.22 | 0.21 | 0.20 | 0.24 | 0.20 | 0.22 |
| F1 Score | 0.07 | 0.00 | 0.03 | 0.23 | 0.02 | 0.11 | 0.02 | **0.24** |
| **Intersection (Valence 4)** | | | | | | | | |
| Precision | 0.12 | 0.01 | 0.03 | **0.46** | 0.03 | 0.15 | 0.04 | 0.42 |
| Recall | 0.24 | **0.34** | 0.23 | 0.25 | 0.22 | 0.28 | 0.23 | 0.25 |
| F1 Score | 0.16 | 0.01 | 0.05 | **0.32** | 0.05 | 0.20 | 0.07 | 0.31 |
| **Complex (Valence 5+)** | | | | | | | | |
| Precision | 0.45 | 0.33 | 0.39 | 0.80 | 0.35 | 0.53 | 0.33 | **0.82** |
| Recall | 0.87 | **0.96** | 0.90 | 0.90 | 0.90 | 0.90 | 0.90 | 0.90 |
| F1 Score | 0.59 | 0.50 | 0.54 | 0.85 | 0.51 | 0.67 | 0.49 | **0.86** |

