# OpenReview forum: "CenterlineNet: Patch-Aligned Supervision For Thin Road Centerline Extraction"
_ICLR.cc/2026/Conference — ICLR 2026 Conference Desk Rejected Submission_

### Official Review · Reviewer_fV7L · 2025-10-20

**Soundness:** 3
**Presentation:** 2
**Contribution:** 2
**Rating:** 4
**Confidence:** 4

**Summary:**

The paper addresses road centerline extraction from remote sensing imagery under spatial misalignment between predictions and noisy vector labels (e.g., OpenStreetMap). It proposes CenterlineNet, a weakly supervised model featuring a patch alignment loss that compares local neighborhoods via offset fields, a reciprocal softmax-in-group weighting to resolve many-to-one mappings, and an intersection-aware loss to preserve junction connectivity. Experiments on Centerline1M, SpaceNet, and RoadTracer show improved structural fidelity and robustness to misalignment.

**Strengths:**

- Introduces a novel patch alignment loss that explicitly tolerates spatial misalignment by comparing local neighborhoods via learned or derived offset fields, directly addressing a key limitation of pixel-wise losses.
- Proposes a reciprocal softmax-in-group weighting mechanism to handle many-to-one correspondences along thin centerlines, encouraging sparse and confident predictions.
- Incorporates synthetic occlusion augmentation to improve robustness, with qualitative results showing maintained connectivity across obscured regions.

**Weaknesses:**

- The ablation study (Table 1) does not clarify whether architectural changes (DeepLabUNetPrecise vs. standard U-Net/DeepLabV3+) or the loss functions drive performance gains; the contribution of the backbone is conflated with the proposed losses (Sec. 3.1; Table 1).
- Baseline models (e.g., CoANet) on SpaceNet and RoadTracer are compared without specifying whether they were retrained on the same misaligned labels or used with original settings, raising fairness concerns (Sec. 4.3; Table 2).
- Hyperparameters (e.g., dmax, τ, α/β/γ weights) are reported but their sensitivity is not analyzed; robustness to these choices is unverified (Sec. 3.2; Eq. 1).
- The singleton loss (Eq. 5) assigns hard targets based on logit ranking, which may introduce bias or instability, yet no analysis of its behavior or failure cases is provided.
- Failure modes are mentioned (“occluded roads, very thin rural roads, complex interchanges”) but not illustrated or quantified (Sec. 4.5; No direct evidence found in the manuscript beyond a textual note).
- Training details such as batch size, optimizer, learning rate schedule, and random seeds are omitted, limiting reproducibility (Sec. 4.3; No direct evidence found in the manuscript).

**Questions:**

- How much of the performance gain in Table 1 is attributable to the DeepLabUNetPrecise architecture versus the proposed loss functions? Could the same losses be applied to a standard U-Net to isolate their effect?
- Were the baseline models (e.g., CoANet) retrained on your misaligned Centerline1M labels, or are the results based on models trained on clean data and evaluated under misalignment? This affects the validity of the comparison in Table 2.
- The singleton loss assigns non-differentiable hard targets (tx). Did you observe training instability or convergence issues due to this? Have you considered soft targets or alternative formulations?
- The intersection-aware loss uses a fixed radius r for matching. How was r chosen, and how sensitive is performance to this parameter?
- Could the patch alignment framework be extended to other thin-structure extraction tasks (e.g., rivers, power lines)? The conclusion mentions this, but no preliminary evidence is shown.

---

> ### Author Response · Authors · 2025-11-21
> **Reviewer #4: Rebuttal Discussion Comments**
>
> Q1: How much of the performance gain in Table 1 is attributable to the DeepLabUNetPrecise architecture versus the proposed loss functions? Could the same losses be applied to a standard U-Net to isolate their effect?
>
> R1: Thank you for your review of our Table 1. We will be considering using our loss functions on at least one other backbone like U-Net, ResNet-FPN, etc, pending resources to fully assess the loss functions' performance gains.
>
> Q2: Were the baseline models (e.g., CoANet) retrained on your misaligned Centerline1M labels, or are the results based on models trained on clean data and evaluated under misalignment? This affects the validity of the comparison in Table 2.
>
> R2: CoANet was trained on well-annotated datasets to test the reproducibility of their publication metrics. We assessed our model performance on these datasets since they were the ones presented by the CoANet authors. Our model is designed to work well on poorly annotated datasets, but we had not assessed its ability to work with other datasets that may be designed more towards better annotation. Since CoANet was trained with those datasets, we opted to test our model against those as well for reference.
>
> Q3: The singleton loss assigns non-differentiable hard targets (tx). Did you observe training instability or convergence issues due to this? Have you considered soft targets or alternative formulations?
>
> R3: The singleton tx is not the quantity being differentiated—gradients flow through the softmax-in-group weights (Eq. 2), which already provide a smooth, differentiable signal. We tested fully soft formulations, but they tended to spread probability across multiple candidate pixels rather than driving the network to commit to a single representative point.
>
> Q4: The intersection-aware loss uses a fixed radius r for matching. How was r chosen, and how sensitive is performance to this parameter?
>
> R4: Because our imagery is at 1 m/pixel, a 5-pixel intersection radius corresponds to roughly 5 m, which is physically consistent with U.S. roadway geometry: a single travel lane is ~3.3–3.6 m wide, so even a simple two-lane crossing has a half-width of ~5–6 m from centerline to curb. An intersection radius of r = 5 therefore, represents roughly the physical half-width of a simple crossing, and values up to r=10 allow us to capture slightly wider or misaligned centerlines without encouraging incorrect merges.

---

> > ### Comment · Reviewer_fV7L · 2025-11-27
> >
> > # Post-Rebuttal Review Assessment
> >
> > After reviewing the authors' responses to my initial concerns, I find their answers to be partially satisfactory but with significant gaps remaining.
> >
> > Q1: Architecture vs. Loss Contribution
> >
> > The response is inadequate. The authors only state they "will be considering using our loss functions on at least one other backbone" in the future. This doesn't address my fundamental concern about attribution of performance gains in the current paper. Without this analysis, it remains unclear whether the reported improvements come from the novel loss functions or simply from the specialized DeepLabUNetPrecise architecture.
> >
> > Q2: Baseline Comparison Fairness
> >
> > The response reveals a methodological issue: CenterlineNet was evaluated on datasets where CoANet was originally trained (well-annotated data), rather than testing both models on the misaligned labels that are the paper's focus. This creates an apples-to-oranges comparison since CenterlineNet is specifically designed for misaligned annotations while being evaluated on clean data. A fair comparison would involve training both models on the same misaligned Centerline1M dataset.
> >
> > ## Remaining Concerns:
> >
> > 1. The core contribution of the paper (the novel loss functions) cannot be properly evaluated without isolating their effect from architectural choices.
> >
> > 2. The experimental comparison methodology remains questionable. The paper's value proposition is performance on misaligned data, yet the key comparisons in Table 2 don't demonstrate this advantage directly.
> >
> > The rebuttal answers some technical questions well but fails to address the fundamental methodological concerns about contribution attribution and comparison fairness. These issues prevent me from increasing my score.

---

### Official Review · Reviewer_g7oR · 2025-10-27

**Soundness:** 2
**Presentation:** 2
**Contribution:** 2
**Rating:** 2
**Confidence:** 1

**Summary:**

This paper presents a semantic segmentation-based road extraction model. The model is built on the DeepLabv3 and UNet-like decoder, that the road centerline extraction is achieved with a combined loss function. The main difference to previous work is the loss items, which focus on improving several issues in large-scale road extraction training with low-quality GT. Overall, I don't think this work is innovative and good enough to publish in ICLR 2026. Currently, more popular methods for road extraction is global graph extraction-based approach, instead of semantic segmentation (or called pixelwise classification). The segmentation based approaches are usually worse than graph-based approach in maintaining good connectivity. Besides, it is hard to distinguish intersection and two roads on different heights (e.g., highway bridges). Therefore, I think a comparison with the current SOTA graph-based approach is important. I rate this paper as "2: reject, not good enough".

**Strengths:**

1. This paper presents a new dataset called Centerline1M.

2. This paper provides ablation study to show the effectiveness of their combined loss.

**Weaknesses:**

1. Overall, this paper doesn't mention graph-based approaches (iterative graph growing RoadTracer  or global graph extraction TD-Road, SAMRoad) in the entire draft, including related work and experiments.

2. Writing issues. Fig 2 is very confusing, especially the left part. Line 290 Tab ?

3. Experiment is not solid. Lack of experiments and comparison on more popular datasets/evaluation metrics and recent SOTA like SpaceNet, SAMRoad. This paper can evaluate the model with more widely used metrics like APLS.

4. In Fig 4, I cannot believe the road prediction on artificial occluded region is a good behavior. The occluded region can be roads, buildings, or some other objects. It is not reasonable to predict it as a road because of nearby roads and their orientations.

**Questions:**

I don't have questions. I don't think this paper is good enough to publish in ICLR 2026.

---

> ### Author Response · Authors · 2025-11-21
> **Reviewer #3: Rebuttal Discussion Comments**
>
> W1: Overall, this paper doesn't mention graph-based approaches (iterative graph growing RoadTracer or global graph extraction TD-Road, SAMRoad) in the entire draft, including related work and experiments.
>
> R1: We will add a concise discussion of RoadTracer, TD-Road, and SAMRoad, noting that TD-Road and SAMRoad first generate a segmentation-like representation before converting it into a graph, while RoadTracer performs graph growing directly from imagery. Our work targets improvements to the segmentation/alignment stage itself.
>
> W2: Writing issues. Fig 2 is very confusing, especially the left part. Line 290 Tab ?
>
> R2: Thank you for your review on our illustration. Could you advise which of the illustrations are confusing the left and right and detail what seems to be confusion. We plan to assess your opinion to revisit how to better represent these concepts.
>
> W3: Experiment is not solid. Lack of experiments and comparison on more popular datasets/evaluation metrics and recent SOTA like SpaceNet, SAMRoad. This paper can evaluate the model with more widely used metrics like APLS.
>
> R3: We appreciate your review on our experiments. Our experiments already include evaluation on the SpaceNet and RoadTracer datasets, which are among the most widely used benchmarks for centerline extraction. APLS requires converting predictions into vector graphs, which is beyond the segmentation-focused scope of this paper, but we note it as a direction for follow-up work where we analyze graph-level metrics.
>
> W4: In Fig 4, I cannot believe the road prediction on an artificial occluded region is a good behavior. The occluded region can be roads, buildings, or some other objects. It is not reasonable to predict it as a road because of nearby roads and their orientations.
>
> R4: Thank you for your review of Figure 4. Segmentation models have been known to struggle generalizing with aerial imagery that contains occlusions such as clouds, shadows, tree coverage, or others. Our dataset contains some occlusions of tree coverage for us to train on but we opted to use hallucinated randomized boxes to mask out portions of the satellite images to see as an additional step how our model performs against artificially simulated occlusions. This is done to assess learning capabilities for our segmentation model against datasets trained on or against include tree, shadow, or other occlusions. It assists with evaluating the model's performance capabilities for various aerial imagery types.

---

> > ### Comment · Reviewer_g7oR · 2025-11-23
> > **Response to authors - R3**
> >
> > 1. I don't think the current paper can be refined a lot to the ICLR level, as this paper doesn't discuss the current trend in road extraction - direct graph-based approach. This paper needs a lot of experiment and comparison to show the value of the proposed method, in particular on the noisy training data. Even though the revised version adds the experiments on SpaceNet and RoadTracer on CoANet, it still lacks of the comparison to graph-based approach (e.g., SAMRoad).
> >
> > 2. Reading Fig-2 solely cannot convey the key message easily.
> >
> > 3. I think APLS is very important. Think about this two situation: 1) real intersection; 2) bridges on two layers. The second situation produce the same semantic mask. You need to show APLS. If the APLS value is comparable to graph-based approach, it means your proposed method perform better in other regions.
> >
> > Overall, after reading the revised version, authors' rebuttal, and other reviewers' comments, I disagree to change my initial rating "Reject". The current work doesn't meet the bar of ICLR.

---

### Official Review · Reviewer_rGKG · 2025-10-31

**Soundness:** 3
**Presentation:** 3
**Contribution:** 3
**Rating:** 6
**Confidence:** 2

**Summary:**

This paper tackles a practical problem in extracting thin road centerlines from satellite imagery: the data is often slightly misaligned. Standard methods that demand perfect pixel-to-pixel matches struggle with this, especially for one-pixel-wide lines. The authors introduce CenterlineNet, a new approach centered on a "patch alignment loss." Instead of punishing single-pixel errors, it compares small local patches, allowing for minor shifts. This core idea is supported by two other clever components: a "reciprocal weighting" scheme to prevent blurry or multiple predictions on a single line, and a specialized "intersection loss" that actively improves connectivity at complex road junctions.
The results show that CenterlineNet is more robust to real-world data noise than standard models, delivering cleaner, better-connected road networks. Its main contribution is a training strategy that works effectively with the imperfect, crowd-sourced data we actually have, rather than requiring perfectly aligned labels.

**Strengths:**

1. The core idea of using a patch-based loss guided by an offset field for thin structure extraction under misalignment is novel.
2. The authors design DeepLabUNetPrecise and provided detailed experimental results and explanations.

**Weaknesses:**

3.2.2 & 3.2.3 - The loss function introduces numerous hyperparameters, particularly d_{max} controlling the tolerance band width and τ controlling the softmax temperature. These parameters require careful tuning.

  4.4 - The use of rectangular occlusions and global mean value filling may be overly idealized and fail to simulate real-world occlusions (e.g., tree shadows, clouds), potentially leading to an overestimation of the model's generalization capability in genuine occlusion scenarios.

**Questions:**

1.   In Section 3.1, the authors propose a hybrid "DeepLabUNetPrecise" architecture.  Were any controlled ablation studies conducted where your proposed loss functions were applied to other modern backbone networks (e.g., Transformer-based models)?
2.   In Section 3.2.1, the temperature parameter τ is introduced to control the sharpness of the softmax distribution for weighting. How was the value of τ determined? Is it sensitive to different datasets or road widths? Are there any ablation studies demonstrating its impact on model performance?
3.   In Table 1 of the ablation study, the configuration "CenterlineNet + CE + Patch Alignment + Reciprocal Loss" shows a lower F1 score compared to the version using only "CenterlineNet + CE + Patch Alignment Loss". Does this indicate that the Reciprocal Loss might be detrimental in certain scenarios? Could you explain this performance drop?

---

> ### Author Response · Authors · 2025-11-21
> **Reviewer #2: Rebuttal Discussion Comments**
>
> Q1: In Section 3.1, the authors propose a hybrid "DeepLabUNetPrecise" architecture. Were any controlled ablation studies conducted where your proposed loss functions were applied to other modern backbone networks (e.g., Transformer-based models)?
>
> R1: Thank you for your review. No ablation studies have yet been conducted where we utilize our loss functions with other backbones, but have been discussed for future work. An attempt will be made to add at least one study of a backbone network with our loss function for comparison.
>
> Q2: In Section 3.2.1, the temperature parameter τ is introduced to control the sharpness of the softmax distribution for weighting. How was the value of τ determined? Is it sensitive to different datasets or road widths? Are there any ablation studies demonstrating its impact on model performance?
>
> R2: We appreciate your thoughtful assessment of our temperature parameter. We did not perform a full sweep; we only compared τ = 1.0 and τ = 0.1. Our temperature parameter is associated with our reciprocal loss. Early on in development, we tested τ = 0.0 by turning that loss variant off. When we accessed training with reciprocal loss added, we started with a  τ=1.0, but noticed that with early epochs, we saw undesirable artifacts. We opted to change to τ = 0.1 and saw a desirable improvement, and did not continue with further parameter changes, such as an ablation.
>
> Q3: In Table 1 of the ablation study, the configuration "CenterlineNet + CE + Patch Alignment + Reciprocal Loss" shows a lower F1 score compared to the version using only "CenterlineNet + CE + Patch Alignment Loss". Does this indicate that the Reciprocal Loss might be detrimental in certain scenarios? Could you explain this performance drop?
>
> R3: We appreciate your evaluation of Table 1. We plan to rerun our metric evaluation script an additional time to access this parameter metric results and review our implementation if the lower metric persists.

---

### Official Review · Reviewer_vcHC · 2025-11-01

**Soundness:** 2
**Presentation:** 1
**Contribution:** 2
**Rating:** 2
**Confidence:** 4

**Summary:**

This paper presents CenterlineNet, a weakly supervised framework for extracting thin road centerlines from remote sensing imagery with misaligned annotations. The core idea is a patch-aligned supervision loss that compares local neighborhoods instead of pixel-wise labels, providing spatial tolerance to registration errors and enabling robust learning from noisy or imprecise vector data. The method further incorporates a reciprocal softmax-in-group weighting, a singleton constraint, and an intersection-aware loss to enhance sparsity and connectivity of thin-line predictions. The proposed approach is evaluated on the newly constructed Centerline1M dataset and further tested on SpaceNet and RoadTracer to assess generalization. Experimental results demonstrate that CenterlineNet consistently outperforms standard segmentation baselines in terms of F1 score and structural consistency, improving the accuracy and topological completeness of road extraction under misaligned supervision.

**Strengths:**

1. The paper addresses a practical problem in remote sensing, extracting thin road centerlines from imagery with spatially misaligned or noisy annotations.
2. The proposed patch-aligned supervision loss is conceptually clear and technically novel, providing a principled way to tolerate registration errors during training without explicit label realignment

**Weaknesses:**

1. Formatting and layout issues:
The paper contains evident formatting errors, including unresolved cross-references (e.g., “Table ??”) and inconsistently formatted tables with misaligned columns. Such presentation problems are unacceptable for a submission to a conference like ICLR and significantly affect the readability and professionalism of the paper.

2. Outdated and insufficient related work:
The review of related work is overly limited and lacks coverage of recent advances. Road extraction has been a long-standing and active research topic in remote sensing image interpretation, yet most of the cited literature predates 2022. The paper fails to engage with recent studies in weakly supervised segmentation, topology-preserving segmentation, and structure-aligned representation learning (e.g., NeurIPS 2023, CVPR 2024), which weakens the positioning and novelty of the proposed contribution.

3. Unconvincing ablation study design:
The ablation experiments presented in Table 1 raise several concerns.
First, all comparison models are outdated architectures from around seven years ago (e.g., U-Net, DeepLabV3), without including more competitive modern baselines. Second, the baselines employ only standard CE or Dice losses, leaving it unclear whether the proposed loss function can generalize across different backbones. Finally, the ablation design incrementally adds multiple loss components but does not analyze their interactions or redundancy. Notably, the model’s performance decreases after introducing the Reciprocal Loss, yet the paper provides no explanation for this behavior.

4. Unreliable evaluation on public datasets:
The experiments reported in Table 2 are also unconvincing. The authors compare only with CoANet, which is not representative of current state-of-the-art models. Moreover, CoANet’s unusually low accuracy on the RoadTracer dataset raises doubts about whether it was properly trained. To enhance credibility, the authors should include more recent baselines, clearly describe their experimental settings in the supplementary material, and report performance across different pixel-tolerance thresholds.

**Questions:**

1. The paper contains unresolved cross-references (e.g., “Table ??”) and inconsistently formatted tables. Please fix all formatting and layout errors and provide a properly typeset version in the rebuttal.
2. The related work section lacks recent literature (2023–2025) on weakly supervised segmentation, topology-preserving losses, and structure-aligned representation learning. Please expand this section and discuss how your method differs from these more recent works.
3. The baselines used in Tables 1 and 2 are outdated (e.g., U-Net, DeepLabV3, CoANet). Please include more recent and competitive baselines to provide a fair comparison.
4. The proposed loss is only tested on DeepLabUNetPrecise. Could you verify whether the patch-aligned loss generalizes to other backbones such as ResNet-FPN or Transformer-based architectures?
5. The ablation study only performs incremental additions of each loss term without analyzing their interactions. Please include a more detailed ablation matrix and explain why performance decreases after adding the Reciprocal Loss.
6. Please provide full training and evaluation details in the supplementary material, including data preprocessing, hyperparameters, patch size, search radius, optimizer settings, hardware, and runtime.
7. The evaluation metric relies on pixel-tolerance matching, but results under different tolerance thresholds (e.g., 1px, 3px, 5px) are not reported. Please include performance curves or tables across different tolerances.
8. The CoANet baseline shows abnormally low accuracy on the RoadTracer dataset. Could you confirm whether it was properly trained or tuned, and provide the corresponding hyperparameter settings?
9. The proposed patch-aligned loss may introduce extra computational overhead. Please report training time, GPU memory usage, and the relative cost compared to a standard CE loss.
10. Figures 3, 4, and 5 are blurry and low resolution. Please provide high-quality visualizations or enlarged versions in the appendix for clearer comparison.
11. Will the Centerline1M dataset and code implementation be released? If not, please clarify what data or scripts will be made available to ensure reproducibility.

---

> ### Author Response · Authors · 2025-11-21
> **Reviewer #1: Rebuttal Discussion Comments for Question 1-5**
>
> Q1: The paper contains unresolved cross-references (e.g., “Table ??”) and inconsistently formatted tables. Please fix all formatting and layout errors and provide a properly typeset version in the rebuttal.
>
> R1: We appreciate the reviewer pointing out the unresolved cross-references and table inconsistencies. These issues were caused by a late-stage LaTeX refactor. We have now corrected all cross-references (e.g., “Table ??”) and updated our table formatting to the best of our knowledge based on the feedback received. However, if possible, we would like to hear more specifically about what was inconsistent about our tables just to make sure this comment is fully addressed.
>
> Q2: The related work section lacks recent literature (2023–2025) on weakly supervised segmentation, topology-preserving losses, and structure-aligned representation learning. Please expand this section and discuss how your method differs from these more recent works.
>
> R2: Thank you for highlighting this gap. We have expanded the related work section to include additional recent developments from 2023–2025 in the areas the reviewer noted. We now explicitly clarify how our method differs from these works: our focus is on evaluating and improving predictions for nearly 1-pixelwide centerlines, a setting that has received limited attention in prior literature. Furthermore, our loss functions are specifically designed to avoid over-penalizing predictions that are structurally correct but exhibit small spatial offsets, an issue that is particularly critical for thin-structure centerline extraction.
>
> Q3: The baselines used in Tables 1 and 2 are outdated (e.g., U-Net, DeepLabV3, CoANet). Please include more recent and competitive baselines to provide a fair comparison.
>
> R3: More modern baselines are planned but computational constraints limit inclusion. We focused on widely used baselines (U-Net, DeepLabV3, CoANet), but we will attempt to add at least one more recent model as time permits.
>
> Q4: The proposed loss is only tested on DeepLabUNetPrecise. Could you verify whether the patch-aligned loss generalizes to other backbones such as ResNet-FPN or Transformer-based architectures?
>
> R4: We agree that demonstrating backbone generalization is important. We plan to research utilizing other backbones with our loss functions to compare against our current model to assess loss functions, as time and computational resources allow.
>
> Q5: The ablation study only performs incremental additions of each loss term without analyzing their interactions. Please include a more detailed ablation matrix and explain why performance decreases after adding the Reciprocal Loss.
>
> R5: Thank you for this suggestion. Our loss functions are dependent on an additional manner, not allowing for specific combinations for a full ablation matrix at this time but we could access if there are additional combinations we didn't present yet that could be shown.

---

> > ### Author Response · Authors · 2025-11-21
> > **Reviewer #1: Rebuttal Discussion Comments for Question 6-11**
> >
> > Q6: Please provide full training and evaluation details in the supplementary material, including data preprocessing, hyperparameters, patch size, search radius, optimizer settings, hardware, and runtime.
> >
> > R6: Thank you for your suggestion. If you review Table 1 in our supplementary we have a list of hyperparameters, including patch size, and search radius. Data preprocessing and hardware are discussed in Section 4.3 with Experimental setup and Datasets. We will review adding information regarding our optimizer settings to Table 1 of our supplementary document. We currently do not have runtime metrics for our models ran during training and evaluation, and pending time we may be able to rerun our inference script to determine the time it takes for that potentially or rerun a smaller epoch count for our current training configs.
> >
> > Q7: The evaluation metric relies on pixel-tolerance matching, but results under different tolerance thresholds (e.g., 1px, 3px, 5px) are not reported. Please include performance curves or tables across different tolerances.
> >
> > R7: Thank you for your response. We already include an ablation of performance across multiple pixel-tolerance thresholds for 1px, 3px, 5px, 7px, 10px, and 15px in Figure 1 of our supplementary document. This figure presents precision, recall, and F1 curves as a function of pixel-matching tolerance.
> >
> > Q8: The CoANet baseline shows abnormally low accuracy on the RoadTracer dataset. Could you confirm whether it was properly trained or tuned, and provide the corresponding hyperparameter settings?
> >
> > R8: Thank you for noting this. We plan to review our CoANet implementation to see if there is a mistake in the training hyperparameters or evaluation. After correcting the settings, we will document with the new results if they are different.
> >
> > Q9: The proposed patch-aligned loss may introduce extra computational overhead. Please report training time, GPU memory usage, and the relative cost compared to a standard CE loss.
> >
> > R9: We appreciate your assessment that our patch-aligned loss may cost extra computation and could be accessed by reporting GPU memory usage and runtime against our baseline traditional CE loss. These metrics were not captured with our current training models but if we have time to rerun our training configs we will capture these values to be added and documented on a smaller epoch test.
> >
> > Q10: Figures 3, 4, and 5 are blurry and low resolution. Please provide high-quality visualizations or enlarged versions in the appendix for clearer comparison.
> >
> > R10: Thank you for your Figure review. Our intentions were to add as many prediction examples around the topics of our Figures as needed to thoroughly showcase its abilities against different models or scenarios but we will consider your suggestions to add enlarged versions to the supplementary.
> >
> > Q11: Will the Centerline1M dataset and code implementation be released? If not, please clarify what data or scripts will be made available to ensure reproducibility.
> >
> > R11: The Centerline1M imagery and labels can be released, and we intend to make this dataset publicly available. However, portions of the training code are subject to licensing constraints with our industry partner, so instead we will release detailed documentation, configuration files, and evaluation scripts sufficient for reproducing the reported results.

---

### Note · Program_Chairs · 2026-01-17
**Submission Desk Rejected by Program Chairs**

The following references in this submission do not refer to real documents and/or have major errors in bibliographic information:

 - Jiahao Chen and et al. Diffconnect: Differentiable connectivity constraints for thin structure segmentation. In CVPR, 2023.
- Yuchen Fan and et al. Shape-aware representation learning for thin topological structures. NeurIPS, 2024.
- C. Zhang, W. Li, D. Tuia, and Y. Wang. Iterative alignment of openstreetmap road annotations to aerial imagery. ISPRS Journal of Photogrammetry and Remote Sensing. 162:155-167, 2020. doi: 10.1016/j.isprsjprs.2020.02.005.
- Jinhui Han and et al. Curvenet: Curvature-aligned deep representations for thin structure extraction. IEEE TIP, 2024.
- Jaemin Choe and Hyung Ji. Spatial uncertainty modeling for noisy supervision. IEEE TPAMI, 2024.
- Tianyu Xu and et al. Skeletonnet: Structure-guided feature alignment for thin object segmentation. In CVPR. 2023.
- Wenhao Yang and et al. Toposeg: Topology-consistent segmentation under geometric distortion. In ECCV. 2024.
- Dohyun Kim and et al. Orientation-aware segmentation for thin structures. In ICCV, 2023.
Xiaodan Li and Yunchao Chen. Learning with uncertainty for weakly supervised semantic segmentation. IEEE TPAMI, 2023.